# Complementary mechanisms create direction selectivity in the fly

Juergen Haag[1][†], Alexander Arenz[1][†], Etienne Serbe[1], Fabrizio Gabbiani[2], Alexander Borst[1]*

[1]Max Planck Institute of Neurobiology, Martinsried, Germany; [2]Baylor College of Medicine, Houston, United States

**Abstract** How neurons become sensitive to the direction of visual motion represents a classic example of neural computation. Two alternative mechanisms have been discussed in the literature so far: preferred direction enhancement, by which responses are amplified when stimuli move along the preferred direction of the cell, and null direction suppression, where one signal inhibits the response to the subsequent one when stimuli move along the opposite, i.e. null direction. Along the processing chain in the *Drosophila* optic lobe, directional responses first appear in T4 and T5 cells. Visually stimulating sequences of individual columns in the optic lobe with a telescope while recording from single T4 neurons, we find both mechanisms at work implemented in different sub-regions of the receptive field. This finding explains the high degree of directional selectivity found already in the fly's primary motion-sensing neurons and marks an important step in our understanding of elementary motion detection.

*For correspondence: borst@ neuro.mpg.de

[†]These authors contributed equally to this work

## Introduction

Flies see the world through a hexagonal array of facets each equipped with its own small lens focusing the light onto 8 photoreceptors. Photoreceptors send their axons into the optic lobe, which consists of four consecutive layers of neuropil called lamina, medulla, lobula and lobula plate (*Figure 1a*). Each neuropil layer is made up of retinotopically organized columns together containing roughly 100 different neurons per column (*Fischbach and Dittrich, 1989*). Within the optic lobe, visual motion information is extracted in parallel pathways encoding light increments (ON) and decrements (OFF) (*Joesch et al., 2010*; *Eichner et al., 2011*; *Joesch et al., 2013*). Both pathways bifurcate in the lamina and lead, via a set of specific medulla interneurons, onto the dendrites of T4 and T5 cells, respectively (*Takemura et al., 2013*; *Shinomiya et al., 2014*). Both T4 and T5 cells exist in 4 subgroups tuned to one of the four cardinal directions of motion and project into four layers of the lobula plate (*Maisak et al., 2013*) (*Figure 1b*). There, they form monosynaptic cholinergic excitatory connections with the dendrites of the large-field lobula plate tangential cells (*Mauss et al., 2014*; *Schnell et al., 2012*) as well as with lobula plate intrinsic neurons. These intrinsic neurons, in turn, inhibit tangential cells in the adjacent layer (*Mauss et al., 2015*). Since none of the neurons upstream from T4 and T5 respond to visual motion in a direction-selective way (*Behnia et al., 2014*; *Strother et al., 2014*; *Meier et al., 2014*; *Ammer et al., 2015*; *Serbe et al., 2016*), T4 and T5 cells are the first neurons in the processing chain where directional information is represented explicitly (*Maisak et al., 2013*; *Fisher et al., 2015*).

To investigate the mechanism leading to direction selectivity in T4 cells, we applied apparent motion stimuli where, instead of continuously moving (*Figure 1c*, top), a bar is abruptly stepped from one location to a neighboring one (*Figure 1c*, bottom). These stimuli lend themselves well to discriminate between preferred direction ('PD') enhancement (*Hassenstein and Reichardt, 1956*) (*Figure 1d*, left) and null direction ('ND') suppression (*Barlow and Levick, 1965*) (*Figure 1d*, right)

**eLife digest** The brain extracts information from signals delivered from the eyes and other sensory organs in order to direct behavior. Understanding how the interactions and wiring of a multitude of individual nerve cells process and transmit this critical information to the brain is a fundamental goal in the field of neuroscience.

One question many neuroscientists have tried to understand is how nerve cells in an animal's brain detect direction when an animal sees movement of some kind – so-called motion vision. The raw signal from the light receptors in the eye does not discriminate whether the light moves in one direction or the other. So, the nerve cells in the brain must somehow compute the direction of movement based on the information relayed by the eye.

For more than half a century, major debates have revolved around two rival models that could explain how motion vision works. Both models could in principle lead to neurons that prefer images moving in one direction over images moving in the opposite direction – so-called direction selectivity. In both models, the information about the changing light levels hitting two light-sensitive cells at two points on the eye are compared across time. In one model, signals from images moving in a cell's preferred direction become amplified. In the other model, signals moving in the unfavored direction become canceled out. However, neither model perfectly explains motion vision.

Now, Haag, Arenz et al. show that both models are partially correct and that the two mechanisms work together to detect motion across the field of vision more accurately. In the experiments, both models were tested in tiny fruit flies by measuring the activity of the first nerve cells that respond to the direction of visual motion. While each mechanism alone only produces a fairly weak and error-prone signal of direction, together the two mechanisms produce a stronger and more precise directional signal. Further research is now needed to determine which individual neurons amplify or cancel the signals to achieve such a high degree of direction selectivity.

as the response to the sequence can be compared with the sum of the responses to the luminance pulses given in isolation ('linear expectation'): in case of preferred direction enhancement, the response to the sequence along the preferred direction is larger than the linear sum of responses to the isolated pulses and identical if the sequence is along the null direction; In case of null direction suppression, the response to the sequence along the preferred direction is identical to the linear sum of responses to the isolated pulses and smaller if the sequence is along the null direction.

The columnar organization of the optic lobes stretches from the laminar cartridges through the medulla and into the lobula and lobula plate. Individual T4 and T5 neurons extend their dendrites across multiple columns (*Figure 1e*) and receive synaptic inputs from different medulla cell types located in different columns relative to the home columns of the T4 or T5 neuron and to each other (*Takemura et al., 2013*; *Shinomiya et al.; 2014*). To understand the particular contribution of those inputs it seemed necessary to precisely place the stimuli onto the columnar raster of the fly's optic lobe.

The structure of the optical system and of the neuronal wiring in flies obeys the neural superposition principle (*Kirschfeld, 1967*; *Braitenberg, 1967*). Those photoreceptors R1-6 from 6 neighboring ommatidia that share the same optical axis converge on the same lamina cartridge, thus forming an optical column that represents the unit of spatial resolution ('neuro-ommatidium'). To visually stimulate these neuro-ommatidia precisely and in isolation, stimulation must consist of parallel rays at angles along the optical axes of those photoreceptors and aligned to the columnar raster. For this, we adopted a telescopic stimulation device (*Franceschini, 1975*; *Schuling et al., 1989*) (*Figure 1f*). As the fly rhabdomeric photoreceptors work as light guides, the raster of optical columns can be visualized by shining light from within the head capsule (antidromic illumination) to align the raster of neuro-ommatidia to the stimulation locations on a micro-display with the help of a CMOS camera.

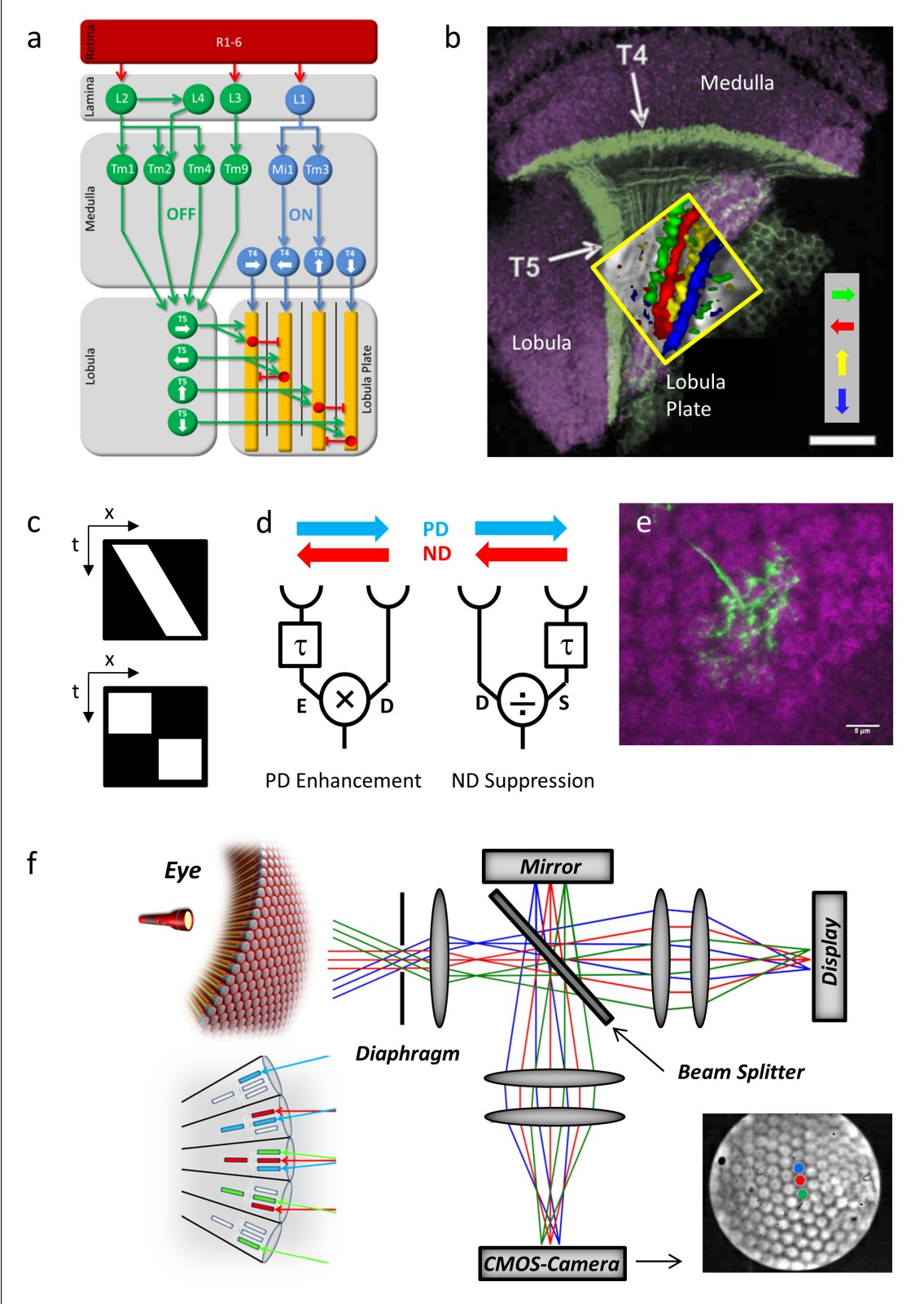

**Figure 1.** Fly optic lobe and visual stimulation. (a) Circuit diagram of the ON and the OFF pathway of fly motion vision. Directionally selective signals are carried via T4 and T5 cells to four layers of the lobula plate, where T4 and T5 cells with the same preferred direction converge on the dendrites of the tangential cells (yellow). Inhibition is conveyed via local interneurons LPi (red). From *Borst and Helmstaedter, 2015*. (b) Confocal image of T4 and T5 cells and their directional tuning. The light green bands indicate the dendrites of T4 and T5 cells. The presynaptic terminals of both T4 and T5 cells

*Figure 1 continued on next page*

*Figure 1 continued*

form four distinct layers within the lobula plate. The inset shows the result of two-photon calcium imaging, revealing four subgroups of T4 and T5 cells tuned to the four cardinal directions. Scale bar, 20 µm. From *Borst and Helmstaedter, 2015*. (c) Continuous (top) versus apparent motion (bottom), shown as x-t-plots where the luminance distribution is shown along one spatial (x) and the time axis (t). During continuous motion at constant velocity, a luminance profile is smoothly drifting along one direction, giving rise to a slanted bar in the x-t-plot. During apparent motion, the luminance profile is stable for a while and then jumps to a new position. (d) Two mechanisms proposed to account for direction selectivity. In each model, the signal from one photoreceptor is delayed by a temporal filter (τ) and fed, together with the direct signal from the neighboring photoreceptor, into a nonlinearity. In case of preferred direction enhancement (left), the delayed signal (E) enhances the direct signal (D), e.g. by a multiplication (E x D), in case of the null direction suppression (right), the delayed signal (S) suppresses the direct signal (D), e.g. by a division (D/S). (e) Immunostaining of a single T4 dendrite in layer 10 of the medulla (green) covering multiple columns (counterstained against bruchpilot, purple). (f) Setup for telescopic stimulation of single lamina columns. Antidromic illumination of the eye (left) results in parallel beams from the 6+2 photoreceptors in neighboring facets with identical optical axes. These are focused in the back focal plane of the objective projected onto a CMOS camera. In addition, an AMOLED display is coupled into the beam path to precisely stimulate single lamina columns. Lower left inset: The fly eye and the principle of neural superposition. Light rays parallel to each other shown in the same color activate different photoreceptors in neighboring ommatidia that converge onto a single column in the lamina ('neuro-ommatidium'). Lower right inset: Picture from the CMOS camera, showing the far field radiation pattern of the *Drosophila* eye. Dot stimuli can be precisely positioned such as to stimulate single lamina columns.

## Results

As a proof of principle, we first expressed the genetically encoded calcium indicator GCaMP6m (*Chen et al., 2013*) in lamina cells L2, recorded from their terminals in the medulla by 2-photon microscopy (*Denk et al., 1990*) and stimulated them with light spots of 1176 ms duration covering a single optical column at various positions. L2 cells responded maximally to stimuli at a certain column, with less than 30% response amplitude to stimuli positioned on surrounding columns, and negligible responses (<12%) to stimuli positioned on columns in the next outer ring (*Figure 2a*). The slight activation of neighboring columns results directly from the optics of the fly's eye, where the visual fields of single ommatidia have Gaussian sensitivity profiles with an acceptance angle roughly matching the inter-ommatidial separation (*Götz, 1965*). These experiments illustrate the specificity of the telescopic stimulation to single lamina cartridges and thus optic lobe column with minimal cross-stimulation of neighboring columns at the physical limit of the fly optical system.

We next used a driver line specific for those T4 and T5 cells sending their axons into layer 3 of the lobula plate that are hence sensitive to upward motion. We recorded from single T4 cells by selecting individual processes in layer three of the lobula plate and confirmed their preference for luminance increments. Repeating the above experiment with T4 cells, we again found maximal responses to the stimulus placed in a single column. Compared to L2, however, the receptive field was found to be larger, with about 50% amplitude to the stimuli onto the six surrounding columns, and roughly 25% to the next outer ring (*Figure 2b*). This indicates that T4 neurons pool excitatory synaptic input from more than one column, consistent with their morphology (*Figure 1e*) and expected from a motion detector that is required to integrate information from spatially offset input.

In order to discriminate between preferred direction enhancement and null direction suppression, we tested T4 cells with three light pulses of 472 ms duration positioned along the dorso-ventral axis of the eye (*Figure 3a*, left). T4 cells responded to the individual pulses with different amplitudes, depending on the position of the stimulus (*Figure 3a*, 'Flicker'; see also *Figure 2b*). When stimulated sequentially from ventral to dorsal (*Figure 3a*, 'Sequence', top middle), the cell responded more strongly (thick blue line) than expected from the sum of the responses to the individual stimuli (thin blue line). The opposite was observed when we stimulated the cell sequentially from dorsal to ventral (*Figure 3a*, 'Sequence', bottom middle): now the cell responded more weakly (thick red line) than expected from the sum of the responses to the individual stimuli (thin red line). We then calculated the nonlinear response component by subtracting the linear expectation from the actual response and found that both preferred direction enhancement (*Figure 3a*, 'Sequence', top right) and null direction suppression (*Figure 3a*, 'Sequence', bottom right) contributes to the directionally selective responses of T4 cells.

We then asked whether these two mechanisms occupied separate or overlapping receptive fields. For that, we presented pulses in four neighboring columns along the dorso-ventral axis, individually (*Figure 3b*, left) as well as sequences of pairs in adjacent columns (*Figure 3b*, middle). Stepping the

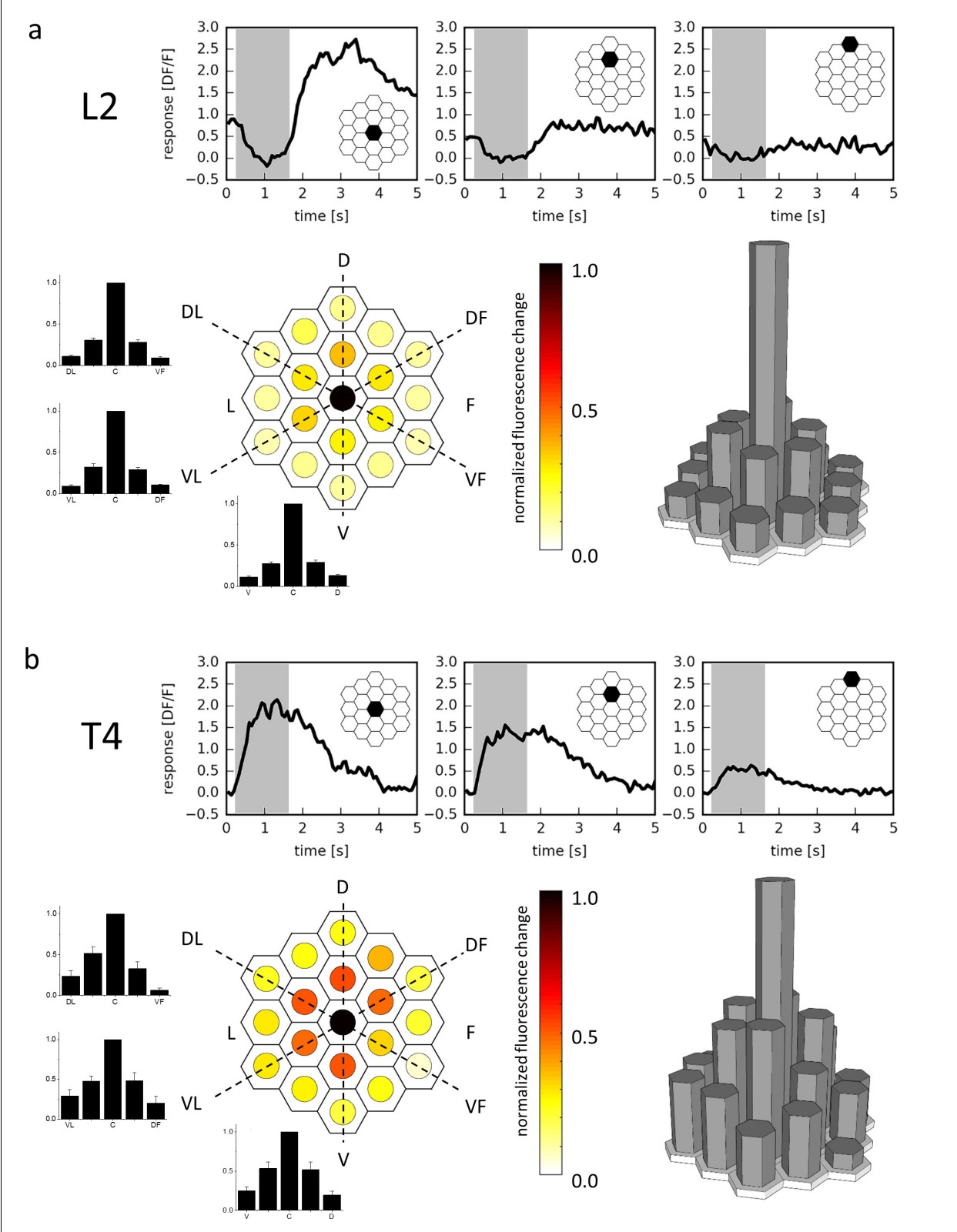

**Figure 2.** Receptive field of L2 (**a**) and T4 (**b**) cells. Three example traces from a single terminal (top, stimulated ommatidium indicated in black) and mean responses (bottom) of L2 cells (**a**, n = 23 cells from six flies) and T4 cells (**b**, n = 10 cells from 10 flies) to flicker stimuli presented at 19 different columnar positions. The responses of individual cells were averaged after alignment to the maximum and normalization and are shown in false color

*Figure 2 continued on next page*

*Figure 2 continued*

code (left) as well as in 3D bar plots (right). In addition, responses are presented as bar plots along the three axes (dashed lines) of the hexagonal array (mean ± SEM). D = dorsal, V = ventral, L = lateral, F = frontal, DL = dorso-lateral, VF = ventro-frontal, VL = ventro-lateral, DF = dorso-frontal.

stimulus up and down between the uppermost columns, we observed only null direction suppression but no preferred direction enhancement (*Figure 3b*, right top). The opposite was true for stimuli at the two lower most locations: here, only preferred direction enhancement was observed, but no null direction suppression (*Figure 3b*, right bottom). For sequences between the two inner locations, both phenomena were visible (*Figure 3b*, right middle). We conclude that the different mechanisms are offset in their receptive fields: preferred direction enhancement is shifted towards the 'null-side', and null direction suppression towards the 'preferred side' with respect to each other.

To assess the time-course of both mechanisms, we presented two pulses of 472 ms duration and delayed one with respect to the other by a variable time-lag (*Figure 3c*). When we placed the pulses onto the ventral part of the receptive field, delaying the upper with respect to the lower one (PD), responses peaked at a delay of about 500 ms (*Figure 3c*, top). When the pulses were placed onto the dorsal part of the receptive field, delaying the lower with respect to the upper one (ND), responses were suppressed and returned to base line at a delay larger than 500 ms (*Figure 3c*, bottom).

We also noticed that this null direction suppression builds up over longer sequences of steps (*Figure 3d*). When the stimulation of the central column was preceded by the stimulation of one directly adjacent column on the 'preferred side', the response to the central pulse was strongly suppressed as before (red trace). This suppression led to a response to the sequence that on average not only falls below the sum of both individual stimulations, but even below the flicker response to the central column (black trace), indicating inhibition. When this null-direction sequence was extended from two to three columns, the resulting response was even smaller (green trace).

The experiments shown in *Figure 3c* also revealed that for zero onset time differences, i.e. simultaneous stimulation of two columns, the response was suppressed as compared to stimulation of just one column. We tested the receptive field of this suppression of the central column systematically by simultaneously stimulating the central column together with one of the columns in the 2 rings surrounding it (*Figure 3e*). We normalized the response to the two stimulated columns with respect to the response to the isolated stimulation of the central column. The resulting response in the T4 neuron was found to be suppressed in comparison to the isolated central column response for simultaneous stimulation of the central column together with another one in the dorsal part of the receptive field.

This suggests that employing apparent motion with flicker stimuli of larger bars should only lead to small flicker responses, possibly occluding null-direction suppression. We tested this prediction by presenting bright horizontal bars of different spatial extent at different locations within the receptive field of individual T4 cells on an LED arena (*Figure 4a,c*). Using a bar size of 180° × 4.5°, indeed, flicker responses of T4 cells were almost undetectable (*Figure 4a*). Apparent motion stimuli along the preferred direction led to pronounced responses, larger than the linear prediction, while null direction sequences did not suppress the negligible sum of flicker responses by a significant amount (*Figure 4b*). This was much different when repeating the same stimuli using bars of only 4.5° × 3° instead: now, as in the experiments employing telescopic stimulation of individual columns, strong flicker responses appeared (*Figure 4c*). Furthermore, in addition to the preferred direction enhancement, pronounced null direction suppression was observed with peak sensitivity within the dorsal part of the receptive field (*Figure 4d*).

What is the virtue of having null direction suppression in addition to the preferred direction enhancement? To address this question, we constructed a simple algorithmic model of a motion detector (*Figure 5a*). In this model, the motion detector receives 3 inputs. A central direct input (D) is flanked by two low-pass filtered inputs: one enhancing input (E) implementing a multiplicative and one suppressing input (S) implementing a divisive non-linearity. This way, the detector is designed to combine a preferred direction enhancement on one and a null direction suppression on the other side within a single motion-sensing unit. Testing the model with single pulses, responses were only detected for positions −1 and 0 (*Figure 5b*, left). Stimulating the model with 2-pulse-

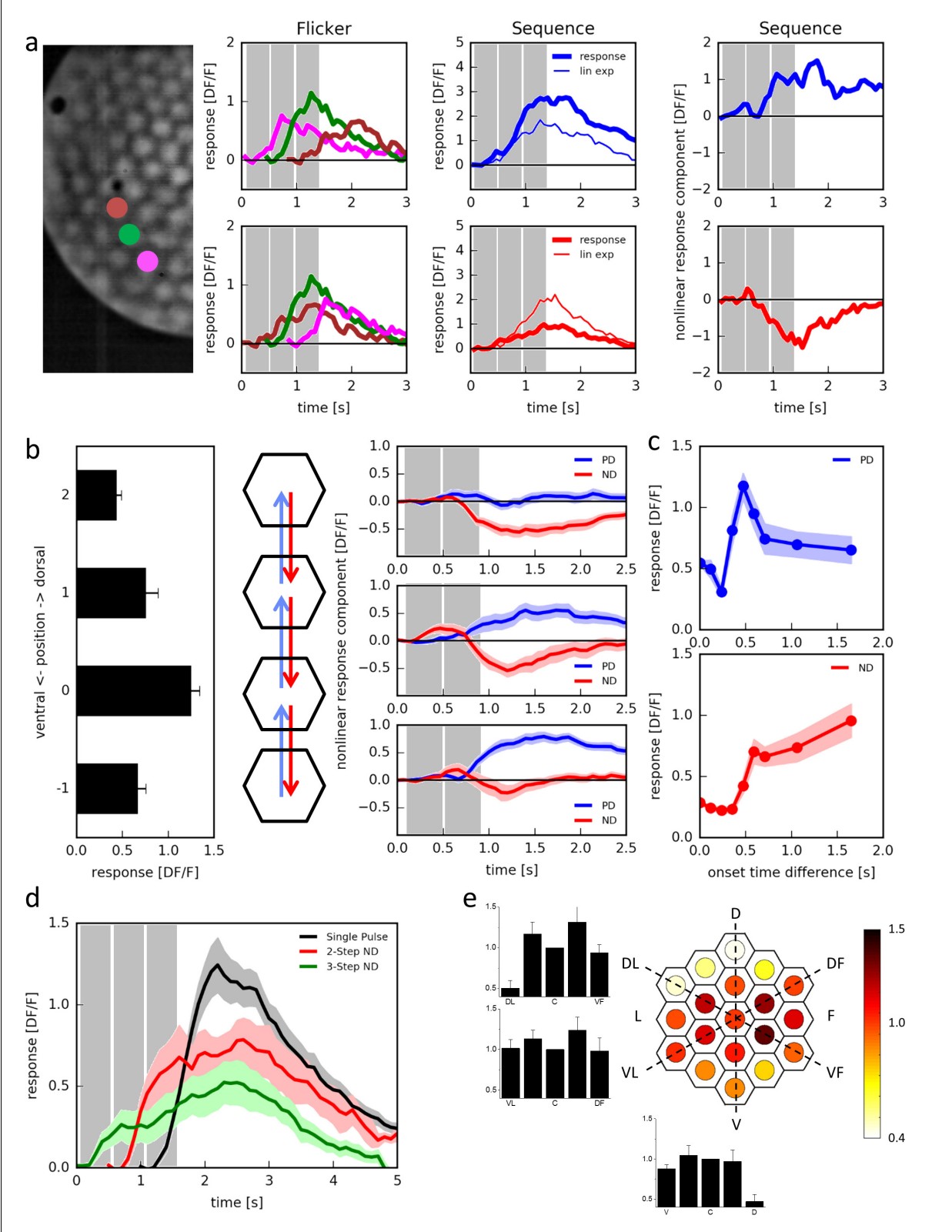

**Figure 3.** Responses of T4 neurons to apparent motion stimulation. (**a**) Response of a single T4 cell recorded in a single sweep to three-step apparent motion stimuli. The image shows the position of the three stimuli. Left: Responses to individual light pulses ('Flicker') delivered at the three different positions. The responses are shifted according to the stimulus protocol used for the subsequent apparent motion stimuli. Middle: Responses of T4 to apparent motion stimuli in preferred and null direction (thick line = measured response, thin line = linear expectation, i.e. sum of responses to the *Figure 3 continued on next page*

*Figure 3 continued*

single light pulses). Right: Nonlinear response component defined as the difference between measured response and linear expectation. The responses are the mean obtained from n = 3 stimulus repetitions. Similar data were obtained in 12 experiments. (**b**) Dependence of the nonlinear two-step response component on the position within the receptive field of T4 cells. Left: Responses to individual light pulses at the four positions. Middle: Stimulus arrangement. Right: Nonlinear response component. Data represent the mean ± SEM (n = 15 cells in 11 flies). (**c**) Responses to 2-step apparent motion stimuli as a function of the onset time delay (top: preferred direction, location −1 -> 0; bottom: null direction, location 2 -> 0). Data represent the mean ± SEM (upper: n = 13 cells from 8 flies; lower: n = 7 cells from 7 flies). (**d**) Average responses to a flicker stimulus of the central neuro-ommatidium (black), to a two-step (red) and a three-step (green) null-direction apparent motion sequence (mean ± SEM, n = 11 T4 cells from eight flies). (**e**) Mean responses of T4 cells to pulses presented to the central column and simultaneously to one of the 18 surrounding columns. Responses were averaged after alignment to the receptive field center, normalized to the flicker response to the central column and are shown in false color code and as bar plots along the three axes of the hexagonal array (mean ± SEM, n = 10 T4 cells from 8 flies). Abbreviations as in *Figure 2*.

sequences at positions −1 and 0 led to pronounced preferred direction enhancement but no null direction suppression (*Figure 5b*, top 2 rows). When the 2-pulse-sequences were delivered at positions 0 and 1, no pronounced preferred direction enhancement occurred, but null direction suppression was substantial (*Figure 5b*, bottom 2 rows). Thus, these simulation results qualitatively match the respective experimental data (compare with *Figure 3a and b*). We next simulated an array of such units and calculated its responses to grating motion at different speeds and directions. We found that the model exhibits a high degree of directional selectivity over a broad range of velocities, with little responses to null direction motion (*Figure 5c*, middle left). In contrast, the model without null direction suppression revealed substantial null direction responses almost as large as its

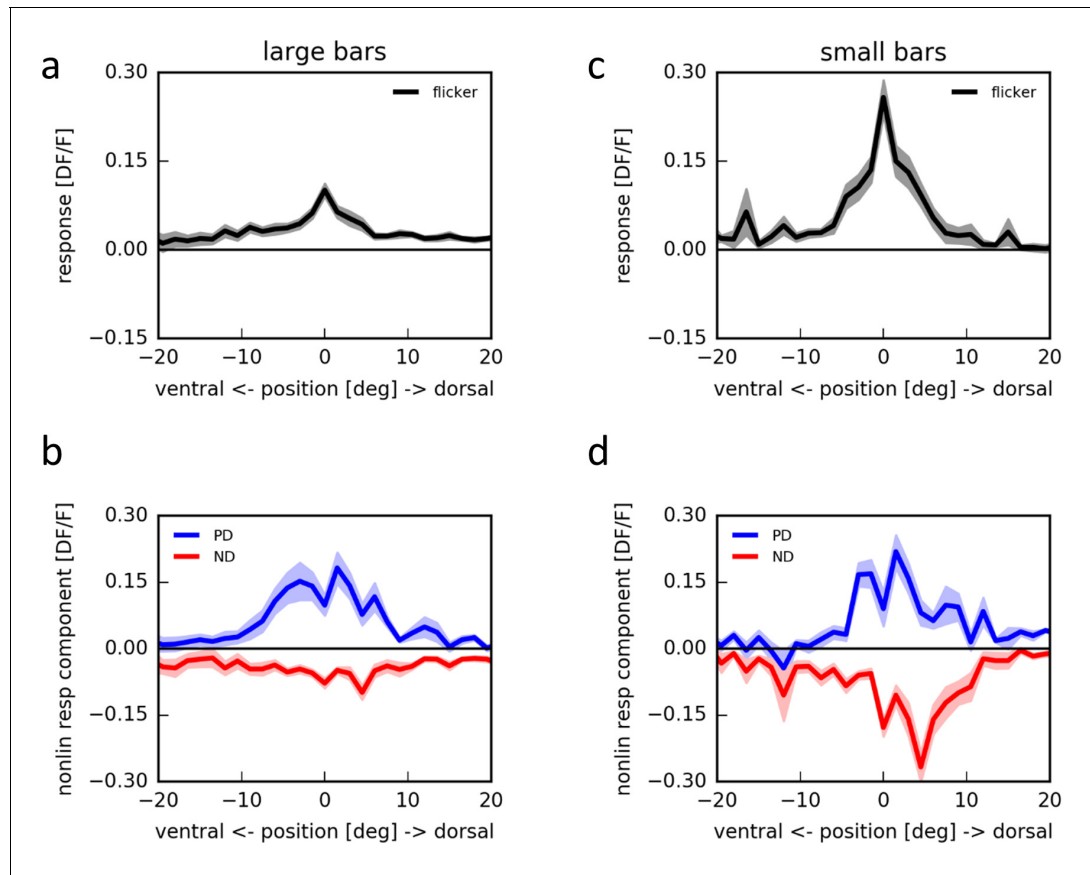

**Figure 4.** Apparent motion stimulation on a LED arena. (**a**) Flicker responses of T4 cells to the presentation of a large horizontal bar (180° × 4.5°) at different elevations on a LED arena, aligned to the elevation evoking the maximum response (n = 17 cells from 5 flies). (**b**) Corresponding non-linear response components to two-step apparent motion sequences at different elevations on an LED arena in the preferred (blue) and null direction (red). (**c,d**) as in (**a,b**) for small horizontal bars (4.5° × 3°) (n = 18 cells from 5 flies).

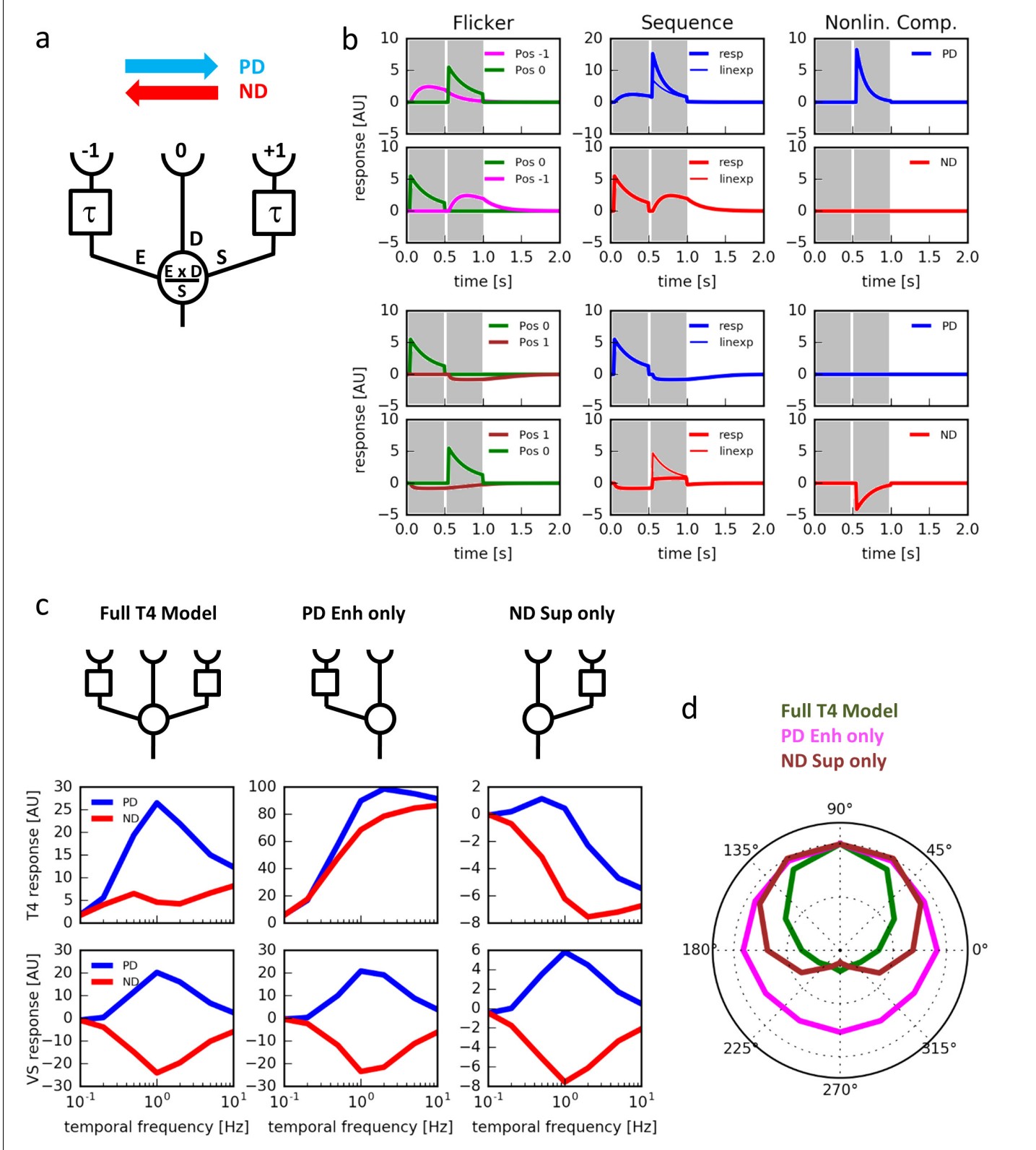

**Figure 5.** Computer simulations. (a) Model T4 cell combining preferred direction enhancement and null direction suppression (compare with *Figure 1d*). (b) Left: Responses to individual light pulses ('Flicker') delivered at the three different positions. The responses are shifted according to the stimulus protocol used for the subsequent apparent motion stimuli. Middle: Responses of the model unit to apparent motion stimuli in preferred and null direction (thick line = measured response, thin line = linear expectation, i.e. sum of responses to the single light pulses). Right: Nonlinear response

*Figure 5 continued on next page*

*Figure 5 continued*

component defined as the difference between measured response and linear expectation. Preferred direction enhancement occurs only between positions −1 and 0, null direction suppression only between positions 0 and 1. (**c**) Comparison of three models (top row): a full model, as in (**a**), implementing preferred direction enhancement and null direction suppression (left), one with preferred direction enhancement only (center) and one with null direction suppression only (right). Temporal frequency tuning of model T4 cells (middle row) and tangential cells (bottom row) using motion of a sine-grating (spatial wavelength = 50°, contrast = 1.0) along the preferred (PD) and null direction (ND), based on those 3 models. (**d**) Directional tuning of an array of model T4 cells using the motion of a sine-grating (spatial wavelength = 50°, contrast = 1.0, temporal frequency = 1.0) for the same 3 models.

preferred direction responses (*Figure 5c*, middle center). The same was true for a model with no preferred direction enhancement (*Figure 5c*, middle right). Interestingly, no such qualitative differences were observed at the level of a model tangential cell, i.e. after subtraction of opponent units (*Figure 5c*, bottom). When we simulated the responses of the model to grating motion into different directions, we found the full model T4 cell to be more sharply tuned to its preferred direction than the one without preferred direction enhancement or the one without null direction suppression (*Figure 5d*). We conclude that combining preferred direction enhancement with null direction suppression leads to a strong direction selectivity of the primary motion-sensing unit.

We next investigated the direction selectivity of T4 and T5 cells by recording their responses to moving gratings presented on an LED arena moving at various speeds and directions. Over a wide range of velocities spanning more than two orders of magnitude, T4/T5 cells responded almost exclusively to upward motion, i.e. along their preferred direction, with little or no responses at all to motion along their null direction (*Figure 6a*). When stimulating the cells with 1 Hz grating motion at various directions, we found a rather sharp directional tuning with about 90° half-width around its preferred direction (*Figure 6b*; see also *Figure 3g* in *Maisak et al., 2013*). To rule out that the highest degree of directional selectivity in T4/T5 cells is achieved by a hitherto unknown reciprocal inhibition between T4/T5 cells with opposite preferred direction, e.g. via inhibitory lobula plate interneurons (*Mauss et al., 2015*) we blocked the synaptic output from all T4 and T5 cells by expression of tetanus toxin light-chain (*Sweeney et al., 1995*). Having confirmed the effectiveness of the block (*Supplementary file 1*), we repeated the above experiments. T4/T5 cells revealed the same high degree of directional selectivity as before (*Figure 6c,d*). From this, we conclude that the mechanism underlying the high degree of directional selectivity in T4 cells does not include the output of oppositely tuned T4 cells but rather originates in the dendrite of the cells.

## Discussion

In the Hassenstein-Reichardt model (*Hassenstein and Reichardt, 1956*) as well as in the Barlow-Levick model (*Barlow and Levick, 1965*) direction selectivity emerges by a nonlinear interaction of asymmetrically filtered signals from adjacent image points: an enhancement along the preferred direction in the first, and a suppression along the null direction in the second model. Both models lead to weakly direction-selective signals in the first place, which, in the Hassenstein-Reichardt model, are improved downstream by subtraction of oppositely tuned components (for review, see *Borst and Helmstaedter, 2015*. Surprisingly, however, a high degree of directional selectivity is found already at the first stage where directional responses are observed, i.e. in T4 and T5 cells (*Maisak et al., 2013*) (*Figure 6*). This can now be explained by the fact that both preferred direction enhancement and null direction suppression are implemented in T4 cells. While the output of the full Hassenstein-Reichardt-correlator after the subtraction stage shows a high degree of directional selectivity in the absence of null-direction suppression, the relatively small differences between large, but poorly tuned signals (see *Figure 5c*, middle panel) would be highly prone to noise. Improving the direction-tuning already at the level of the half-detectors by the additional null-direction suppression increases robustness to noise and might in addition be energetically less costly.

A recent study addressing the mechanisms underlying the elementary motion detectors in *Drosophila* concluded preferred direction enhancement as the sole mechanism (*Fisher et al. 2015*). However, we find that their stimulation paradigm consisting of flashes at 2 positions simultaneously

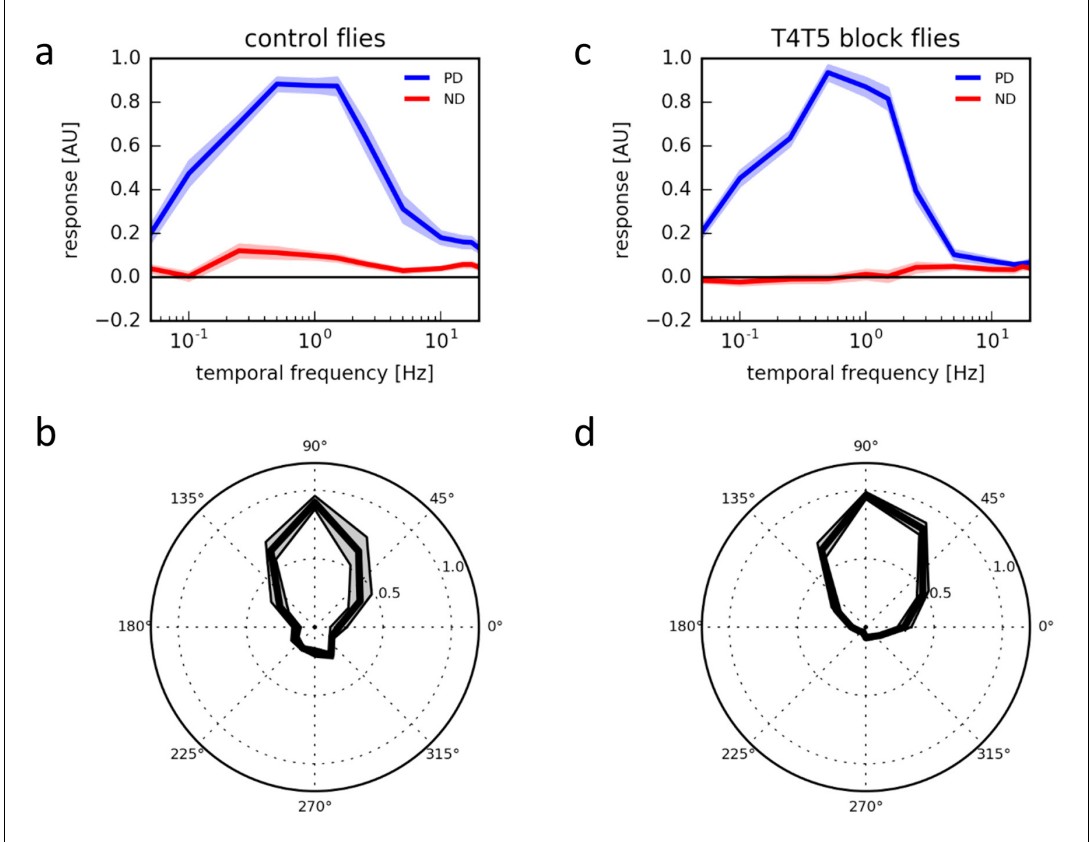

**Figure 6.** Response properties of T4/T5 neurons. (**a**) Temporal frequency tuning of T4/T5 cells in control flies showing the normalized ΔF/F responses to square-wave gratings in the preferred (up, blue) and null direction (down, red) (n = 15 flies). (**b**) Directional tuning of T4/T5 cells to square-wave gratings moving at a temporal frequency of 1 Hz (n = 7). (**c,d**) Temporal (**c**, n = 9) and directional tuning (**d**, n = 9) of T4/T5 cells with their synaptic output blocked by expression of the tetanus toxin light chain. All data represent the mean ± SEM (shaded).

does not allow to discriminate between different mechanisms including lateral inhibition, adaptation and null direction suppression.

Since the operations of preferred direction enhancement and null direction suppression have off-set receptive fields, it requires the cells to collect their input from more than two adjacent image points, in line with anatomical observations (*Figure 1e*; *Takemura et al., 2013*; *Shinomiya et al., 2014*) . In contrast to the cellular implementation of the motion detector proposed by *Behnia et al, 2014* our findings also require that T4 cells receive input from more than two cell types. This again is in agreement with recent connectomic data (for T4 cells: Louis Scheffer, Janelia Research Campus, personal communication; for T5 cells: *Shinomiya et al., 2014*). Finally, functional studies where the synaptic input elements onto T4 and T5 neurons were blocked also suggested the involvement of more than two input elements in the ON- (*Ammer et al., 2015*) as well as in the OFF-pathway (*Serbe et al., 2016*).

Considering the columnar nature of the input elements and the structure of the T4 dendrites covering multiple columns, we predict, that the inputs representing the enhancing, direct and suppressing input segregate on different sub-regions of the T4 dendrite, with enhancing inputs on the 'null-side' and the suppressing inputs on the 'preferred side' of the dendrite. Future experiments will have to map the different cell types of the medulla onto the proposed model (*Figure 5a*) as well as to identify the transmitter receptors, responsible for enhancing, exciting and inhibiting the dendrites of T4 and T5 cells.

## Materials and methods

### Flies

(*Drosophila melanogaster*) were raised at 25°C and 60% humidity on a 12 hr light/12 hr dark cycle on standard cornmeal agar medium. For calcium imaging of T4/T5 cells, flies were used to express the genetically-encoded calcium indicator GCaMP6m (*Chen et al., 2013*) in T4/T5 neurons with axon terminals predominantly in layer 3 of the lobula plate (w⁻; Sp/cyo; VT50384-lexA, lexAop-GCaMP6m/TM6b). For the imaging experiments where synaptic output of T4/T5 activity was blocked, the above flies were crossed to flies driving expression of the tetanus toxin light chain in all T4/T5 cells (w⁻; R59E08-AD / UAS-TNT-E; R42F06-DBD / VT50384-lexA, lexAop-GCaMP6m). For imaging L2 cells, we used the 21D-Gal4 driver line (*Joesch et al., 2010*) to express GCaMP6m.

### Calcium imaging

Fly surgeries were performed, and the neuronal activity was measured from the left optical lobe on a custom-built 2-photon microscope as previously described (*Maisak et al., 2013*). Images were acquired at resolutions between $64 \times 64$ and $256 \times 256$ pixels and frame rates between 1.88 and 15 Hz with the ScanImage software (*Pologruto et al., 2003*) in Matlab.

### Visual stimulation with telescope

Antidromic illumination of the fly's head through the objective used for two-photon-microscopy visualizes the hexagonal mosaic of the optical axes of the ommatidia of a living *Drosophila* (*Franceschini, 1975*; *Schuling et al., 1989*). The far field radiation pattern (FFRP) visible in the back focal plane of the objective (LD Epiplan 50x/0,50, Zeiss) is projected onto a CMOS camera (DCC1545M, Thorlabs) via several lenses, a beam splitter (CM1-BP145B5, Thorlabs) and a diaphragm, to reduce stray light. Visual stimuli are generated on the AMOLED display ($800 \times 600$ pixels, pixel size 15 $\times$ 15 μm, maximal luminance > 1500 Cd/m², lambda = 530 nm; refresh rate 85 Hz) (SVGA050SG, Olightek). Both stimulus pattern and FFRP can be visualized simultaneously by means of the beam splitter and a mirror with the CMOS camera. This allows to precisely position the stimuli onto the FFRP. In order to prevent stimulus light from entering the photomultiplier of the two-photon microscope, light generated by the AMOLED display was filtered with a long pass filter (514 LP, T: 529.4–900 nm, AHF). The AMOLED display was controlled with MATLAB and the psychophysics toolbox (V 3.0.11; *Brainard, 1997*).

### Experimental protocol

We determined the receptive field of T4 cells by stimulating single cartridges with light pulses of 472 ms duration at randomized positions. At each position, three stimulus presentations were delivered. The resulting responses were averaged and the peak of the averaged response was taken. We performed all experiments on T4 cells only. The cells were selected based on their response to light-on stimuli. While T4 cells respond to the onset of a light pulse (*Figure 2b*), the T5 cells respond to the light off. Apparent motion stimuli consisted either of consecutive light stimuli to two or three neighboring cartridges. The second stimulus was presented right after the first turned off, resulting in a delay from onset to onset of 472 ms.

### Visual stimulation with LED arena

The LED arena subtended approximately 180° in azimuth and 90° in elevation with a resolution of 1.5°, based on a design modified from *Reiser and Dickinson (2008)* as previously described (*Maisak et al., 2013*). Stimuli were presented with 3–5 repetitions per experiment in a randomized fashion. All stimuli were presented in full contrast. To measure the directional and temporal frequency tuning, square-wave gratings with a spatial wavelength of 24° spanning the full extent of the stimulus arena were used. For the direction tuning those gratings were moved in 12 directions separated by 30° at a temporal frequency of 1 Hz. To determine the temporal frequency tuning, gratings were moved at temporal frequencies ranging 0.05 to 20 Hz moving in the preferred and null direction. For the apparent motion experiments, either large (180° wide $\times$ 4.5° high) or small (4.5 $\times$ 3°) bright horizontal bars were presented for 400 ms either in isolation (flicker) or in sequences of 2

pulses (apparent motion) offset by 4.5° in the preferred or null direction with a Δt, onset to onset, of 400 ms.

## Data analysis

Data analysis was performed offline using custom-written routines in Matlab. Regions of interests (ROIs) were selected by hand in layer 3 of the lobula plate. The time courses of relative fluorescence changes (ΔF/F) were calculated from the raw imaging sequence. Responses to the stimulus were baseline-subtracted, averaged across repetitions, and quantified as the peak responses over the stimulus epochs. For T4 cells, the baseline was determined during one second before the stimulus, for L2 cells (*Figure 2a*), it was determined during the last second before luminance off-set. Those responses were averaged across experiments. Where indicated, responses were normalized to the maximum average response before averaging. For the apparent motion experiments, non-linear response components were calculated as the differences of the time-courses of the responses to the apparent motion stimuli and the sum of the appropriately time-shifted responses to flicker stimuli at the corresponding positions.

## Model simulations

Visual stimuli were represented as a two-dimensional array (200 x 300) at 1° spatial and 10 ms temporal resolution, mapped onto a linear array of 40 visual columns. The signal amplitude ranged from 0 to 1. The input to the ON pathway, as represented by lamina neuron L1, was modeled as previously described (*Eichner et al., 2011*). Briefly, the local luminance signal was high-pass-filtered (1st order, $\tau$ = 250 ms), and 10% of the DC value was added with subsequent rectification. The T4 cell was then modeled as receiving input from three adjacent columns (*Figure 5a*): an enhancing signal E at position −1, representing the low-pass filtered signal of L1 (1st order, $\tau$ = 250 ms), a direct signal D at position 0 which is identical to L1, and a suppressing signal S again representing the low-pass filtered signal of L1 (1st order, $\tau$ = 250 ms). The response was calculated as the product of E and D, divided by S. Signals had the following weights k: $k_E$ = 5, $k_D$ = 5, $k_S$ = 10. To avoid division by zero and to account for flicker responses, a DC term of 1.0 was added to each signal. To simulate flicker and apparent motion stimuli (*Figure 5b*), light pulses of 450 ms length and amplitude 1.0 were delivered to selected columns and the response of an individual model T4 cell was evaluated. To simulate the responses to sine-gratings (*Figure 5c,d*), responses were calculated either as the summed responses of an array of such units ('T4 response') or after subtracting the signals of oppositely tuned T4 cells ('VS response'). Simulations without null direction suppression or preferred direction enhancement were performed by setting either $k_E$ or $k_S$ to zero. Software was written in the Python programming language and is available as source code.

## Acknowledgements

We are grateful to Wolfgang Essbauer and Romina Kutlesa for fly work, to Stefan Prech for the 3D illustration in *Figure 2*, to Jesus Pujol-Marti for providing us with the picture of an individual T4 cell (*Figure 1e*), to Nicolas Franceschini for advice, to Alex Mauss and Aljoscha Leonhardt for carefully reading the manuscript, and to Winfried Denk for many discussions. This study was funded by the Max-Planck-Society, the Deutsche Forschungsgemeinschaft (SFB 870) and by a fellowship of the Alexander-von-Humboldt Foundation to FG.

## Additional information

### Competing interests

AB: Reviewing editor, *elife*. The other authors declare that no competing interests exist.

### Funding

| Funder | Grant reference number | Author |
| --- | --- | --- |
| Max-Planck-Gesellschaft | | Juergen Haag<br>Alexander Arenz<br>Etienne Serbe |

|  |  | Alexander Borst |
|---|---|---|
| Alexander von Humboldt-Stif-tung |  | Fabrizio Gabbiani |
| Deutsche Forschungsge-meinschaft | SFB 870 | Alexander Borst |

The funders had no role in study design, data collection and interpretation, or the decision to submit the work for publication.

## Author contributions
JH, Designed the experiments and evaluations, Performed and analyzed the experiments using the telescopic stimulation, Conception and design; AA, Performed and analyzed the experiments using the LED arena, Designed the experiments and evaluations, Conception and design; ES, Recorded from tangential cells in T4/T5 block flies; FG, Supported by a fellowship of the Alexander-von-Humboldt Foundation and participated in the early phase by constructing the telescopic stimulus device; AB, Designed the experiments and evaluations, Wrote the manuscript with the help of all authors, Conception and design, Drafting or revising the article

## Additional files

### Supplementary files
• Supplementary file 1. Recordings from tangential cells in flies where synaptic output from all T4 and T5 cells was blocked by expression of tetanus toxin (a,c) and in control flies (b,d). (a,b) Responses of tangential cells of the Vertical System (VS) to square-wave gratings moving in the preferred (blue) and null (red) direction of T4T5 block (A, w-; R59E08-AD / UAS-TNT-E; R42F06-DBD / VT50384-lexA, lexAop-GCaMP6m; n = 5 cells in 3 flies) and control flies (B, w-; R59E08-AD / cyo; R42F06-DBD / VT50384-lexA, lexAop-GCaMP6m; n = 9 cells in 5 flies). (c,d) Directional tuning of the same cells to square-wave gratings moving in 12 different directions. Preferred direction (blue) and null direction (red) responses were averaged over the first second of stimulation. Negative null direction responses were plotted on the opposite polar coordinates. Stimuli were presented on an LED arena (see Materials and methods). Gratings had a spatial wavelength of 24°, a contrast of 1 and moved at 24°/s, i.e. at a temporal frequency of 1 Hz. In all panels, data represent the mean ± SEM. DOI: 10.7554/eLife.17421.009

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
