## [Decision Letter]

Thank you for submitting your article "Complementary Mechanisms Create Direction Selectivity in the Fly" for consideration by *eLife*. Your article has been reviewed by three peer reviewers, one of whom, Fred Rieke, is a member of our Board of Reviewing Editors and the evaluation has been overseen by David Van Essen as the Senior Editor. The following individuals involved in review of your submission have agreed to reveal their identity: Tom Baden (Reviewer #2); Vivek Jayaraman (Reviewer #3).

The reviewers have discussed the reviews with one another and the Reviewing Editor has drafted this decision to help you prepare a revised submission.

This is an interesting and timely paper investigating the mechanisms underlying motion detection in the fly visual system. The paper uses a powerful combination of two-photon calcium imaging and telescopic stimulus delivery, allowing greater spatial control of stimuli than more conventional approaches. All of the reviewers found the work of interest and the experiments generally well done. Several issues came up in the review process: (1) some cases in which the text stated results too strongly given a lack of direct experimental evidence; (2) a need to improve the clarity of the writing and figures. Four specific examples follow, agreed upon as most critical by all three reviewers; others examples are in the individual reviewers comments below.

1) The paper concludes that distinct regions of the T4 dendrites implement different motion detection mechanisms. But there are no direct measures of dendritic signaling in the paper; without such measures, this conclusion should be stated less strongly.

2) The paper suggests a role of inhibition in canceling signals for null direction motion. The evidence for this is based on sublinear summation of signals in the null direction – but this does not uniquely identify inhibition as the underlying mechanism. Was it possible to directly observe inhibition – e.g. to see a decrease in calcium when stimulating a single location in the T4 receptive field with flickering stimuli? Without a direct observation about inhibition this conclusion should also be stated less strongly.

3) The telescopic illumination should be described in more detail. Particularly how effective is it in restricting input spatially? If there is spread beyond the single column targeted, how much could such spread impact the conclusions of the paper? Part of the confusion on this point originates because Figure 1 is difficult to interpret in present form (see point 4 and individual reviewer comments).

4) The figures should be modified to specify exactly what is measured, particularly the optical configuration for each experiment. Specific examples are in the individual reviews below.

Reviewer #1:

This paper describes the origin of directional selectivity in the first cells in the fly visual system that exhibit directional responses. The combination of a telescopic stimulus delivery system to activate specific elements of the motion computation with functional calcium imaging is quite powerful. The results are clear and interesting. However, I found several aspects of the paper difficult to follow. Some specific suggestions to improve clarity follow.

1) Manipulations of stimuli to specific spatial regions are crucial to the paper. But it is often hard to discern exactly what stimulus is used in a given experiment. This could be improved greatly by adding simple schematics to the figures, showing which specific columns are being activated. When those are present (e.g. Figure 2) they help quite a bit. Figure 2 are specific examples in which this could help.

2) The description of Figure 1 is quite minimal and hard to follow. Those figures made much more sense to me after I had read most of the paper, but when they first come up it is not clear what is important about the data, or even how the data was collected. The larger receptive field size of the T4 cells is important; that should get mentioned in the text where this data is described.

3) Figure 3 models. It was not clear to me what the modified models were. Specifically, do the models lacking null direction suppression or preferred direction enhancement still consist of three inputs (as in Figure 3) just with one arm disabled? Or are they models with two inputs only (in a more classic Reichardt configuration)? A schematic of these models added to Figure 3 could be helpful.

4) Figure 3. I am confused by the description of this figure. The legend does not have a panel d, but the legend for panel e suggests it is describing an array of model units, not a full T4 cell. That appears to make sense from comparing Figure 3 and d. But the main text describes a full model T4 cell. Because of these inconsistencies I could not evaluate that part of the paper.

Reviewer #2:

In their MS "Complemetary Mechanisms Create Direction Selectivity in the Fly", Haag et al. use a spatially precise optical setup capable of delivering single-ommatidial stimulation and with 2-photon calcium imaging to probe mechanisms driving selection selectivity in the visual system of the fruit fly. Focussing on T4 cells, the fly-visual system's "ON-DS cells", the authors show that both null-suppression as well as preferred direction enhancement appear to drive directionality in these neurons, computations that likely take place in different dendritic regions of T4s. They follow up their findings in a simple model, which in addition predicts that the 2 mechanisms in tandem yield a sharper spatial tuning than any one mechanism alone. They also demonstrate the "sharp" tuning in T4 cells experimentally (though they do no isolate the 2 mechansims in these experiments – something that I suspect would be quite difficult to do).

Overall, the MS is timely and reports important and to my knowledge novel findings about direction selective visual circuits of the fly. The data is of high quality and the writing is clear and succinct. I only have minor comments

Reviewer #3:

In this tantalizing manuscript, Haag, Borst and colleagues explore the underlying basis of visual motion direction computations in the *Drosophila* optic lobe. In both the vertebrate retina and in the early visual system of flies, this computation has been studied in the framework of two classic models of visual motion direction: the Hassenstein-Reichardt (H-R) model and the Barlow-Levick (B-L) model. Mechanistically, both rely on combining input from spatially offset pathways, with input from an adjacent location temporally delayed relative to the central pathway. One (H-R) produces an excitatory "preferred-direction" (PD) response by multiplication with temporally delayed input from a preceding location, while the other (B-L) uses division by input from a temporally delayed location in the "null direction" (ND). Direct evidence for the latter has been obtained in direction-selective retinal ganglion cells (DSRGCs), where starburst amacrine cells provide the appropriate ND inhibition. In the *Drosophila* optic lobe, the H-R model has thus far been the mechanism most often proposed for direction selectivity for large field motion, with information from "ON" and "OFF" motion pathways represented by so-called T4 and T5 cells respectively being combined in vertical system (VS) and horizontal system (HS) lobula plate tangential cells (LPTCs). Recent anatomical and electron microscopy (EM) evidence has suggested a more complex picture at the level of the input to T4 and T5 neurons – their inputs comprise more cell types than previously believed. In this study, the authors use precise optical stimulation that takes into account the fly's neural superposition eye (Franceschini, 1975) along with two-photon calcium imaging to suggest that B-L-like inhibition may combine with H-R like excitation to shape the responses of T4 neurons. The results are interesting but the evidence provided is indirect and the picture still somewhat murky. Detailed comments follow.

The major claim of the paper is that precise stimulation of single lamina columns reveals PD and ND responses that localize to different sub-regions of the T4 neuron dendrites. However, this claim, which is highlighted in the abstract, goes beyond the evidence presented, which is entirely based on recordings in T4 axons. Unless the authors can present direct anatomical and physiological evidence for such localized dendritic input, I would suggest that this claim be removed or significantly tempered and moved to Discussion.

Similarly, the B-L model proposes inhibition of one signal by another for ND motion, but the authors present no direct evidence for such inhibition. What is shown instead is supra- or sub-linear summation upon sequential (PD vs ND) excitation of successive lamina columns. Why does this have to necessarily result from inhibition in T4 dendrites rather than have presynaptic origins? While it is true that medulla neurons are not known to be sensitive to motion-direction, such neurons are known to have suppressed responses to bars wider than their receptive field (T5 input neurons for OFF pathway; Serbe et al., 2016 from the Borst lab), so might a skewed sampling of such inputs by T4 perhaps confer the observed ND suppression?

Related to the above point, the model (Figure 3) predicts that we should be able to directly observe inhibition (not just suppression of a strong excitatory response), even if it is not large, when the 'wrong' lamina column is illuminated in a flicker experiment. This seems never to be observed in the experiments shown (Figure 1). What is seen instead is sub-linearity (or supra-linearity), which the authors call a non-linear response component. Is this a calcium indicator limitation even with the higher-baseline GCaMP6m? If there is inhibition, might it be easier to pick up directly in the dendrites of single T4 neuron or do the authors find this experimentally infeasible?

In Figure 1, the authors state that their telescopic illumination allows them to illuminate only a single lamina column, but the lamina responses suggest that neighboring columns are also stimulated. In the manuscript (introduction section), these responses are described as 'negligible', but they seem to reach 50% of peak responses. This requires further discussion, because the study's results rely heavily on the high precision of optical illumination. If the responses reflect the limits of the illumination system, it would be useful to discuss how that might affect the interpretation of results. Also, on the same note, it would be great to see what a 'standard' localized stimulus presentation (i.e., targeting single ommatidia and sequences of ommatidia without using the telescopic refinement) would evoke in T4 neurons. This would highlight the importance of telescopic stimulus presentation for visual motion studies, which would be useful to the field.

Although perhaps this is beyond the scope of this study, it would be very interesting to know what T5 neuron responses look like with telescopic stimulation. Do they also show B-L like ND suppression in the OFF pathway? Given that the authors have access to all the T5 inputs (Serbe et al., 2016), this could also provide an easier path to a more complete explanation in these neurons as opposed to T4, where the authors do not say much about what the potential inputs to these neurons might be and do not explore how they might act together to produce the observed response properties.

Lastly, it is still left unclear why T4 (and, perhaps T5) might need a potential B-L like mechanism in addition to H-R if there is no difference between the combination and pure H-R for LPTCs. Perhaps the authors could speculate on this in Discussion.

---

## [Author Response]

1) The paper concludes that distinct regions of the T4 dendrites implement different motion detection mechanisms. But there are no direct measures of dendritic signaling in the paper; without such measures, this conclusion should be stated less strongly.

We recognize that we do not directly image calcium signals originating from the synaptic input within different regions of the dendrites of individual neurons. We rephrased more carefully in terms of “offset receptive fields for preferred-direction enhancement and null-direction suppression"

e.g.: "We conclude that the different mechanisms are offset in their receptive fields: preferred direction enhancement is centered towards the ‘null-side’, and null direction suppression towards the ‘preferred side’ with respect to each other. "

We predict that these offset receptive fields map onto different regions of the dendrite, but moved this argument to the discussion as suggested:

"Considering the columnar nature of the input elements and the structure of the T4 dendrites covering multiple columns, we predict, that the inputs representing the enhancing, direct and suppressing input segregate on different sub-regions of the T4 dendrite, with enhancing inputs on the ‘null-side’ and the suppressing inputs on the ‘preferred side’ of the dendrite. "

2) The paper suggests a role of inhibition in canceling signals for null direction motion. The evidence for this is based on sublinear summation of signals in the null direction – but this does not uniquely identify inhibition as the underlying mechanism. Was it possible to directly observe inhibition – e.g. to see a decrease in calcium when stimulating a single location in the T4 receptive field with flickering stimuli? Without a direct observation about inhibition this conclusion should also be stated less strongly.

We now included additional data to show that apparent motion stimulation in the null direction leads to significantly lower responses than when only the central column was stimulated, and that this suppression builds up with longer stimulation sequences covering more columns in the ND direction (Figure 3). As far as we can see, this can only be explained by inhibition, not by adaptation or similar phenomena. The nature of the inhibition can be shunting or pre-synaptic on the inputs and does not have to be hyperpolarizing. In the former case, it would also not manifest itself as a decrease in the calcium signal.

3) The telescopic illumination should be described in more detail. Particularly how effective is it in restricting input spatially? If there is spread beyond the single column targeted, how much could such spread impact the conclusions of the paper? Part of the confusion on this point originates because Figure 1 is difficult to interpret in present form (see point 4 and individual reviewer comments).

The imaging experiment from L2 neurons illustrate the specificity of the stimulation and the restriction to single columns with minimal overlap into neighboring columns at the optical limit of the fly’s eye. In the meantime, we also increased the number of L2 neurons recorded. We additionally discuss this point now in more detail in the text:

"The slight activation of neighboring columns results directly from the optics of the fly’s eye where the visual fields of single ommatidia have Gaussian sensitivity profiles with an acceptance angle roughly matching the inter-ommatidial separation (Götz, 1965). These experiments illustrate the specificity of the telescopic stimulation to single lamina cartridges and thus optic lobe column with minimal cross-stimulation of neighboring columns."

The corresponding figure was labeled more clearly, including axis labels and a 3D bar plot to immediately visualize the results.

*4) The figures should be modified to specify exactly what is measured, particularly the optical configuration for each experiment. Specific examples are in the individual reviews below.*

We thought a lot about introducing xt-plots throughout, but found that it would add too much of space to the figures. As an example, just the experiment shown in Figure 3 would require the symbols below for visualizing the apparent motion sequences (Figure 7). We, therefore, decided to introduce the apparent motion stimulus in principle in Figure 1 as an xt-plot, and to be as clear as possible in the text otherwise.

Author response image 1.**DOI:**
http://dx.doi.org/10.7554/eLife.17421.010

*Reviewer #1:*

*This paper describes the origin of directional selectivity in the first cells in the fly visual system that exhibit directional responses. The combination of a telescopic stimulus delivery system to activate specific elements of the motion computation with functional calcium imaging is quite powerful. The results are clear and interesting. However, I found several aspects of the paper difficult to follow. Some specific suggestions to improve clarity follow.*

*1) Manipulations of stimuli to specific spatial regions are crucial to the paper. But it is often hard to discern exactly what stimulus is used in a given experiment. This could be improved greatly by adding simple schematics to the figures, showing which specific columns are being activated. When those are present (e.g. Figure 2) they help quite a bit. Figure 2 are specific examples in which this could help.*

We changed the scheme in Figure 2 (now Figure 3) to better illustrate that light pulses were stepped between adjacent neuro-ommatidia.

*2) The description of Figure 1 is quite minimal and hard to follow. Those figures made much more sense to me after I had read most of the paper, but when they first come up it is not clear what is important about the data, or even how the data was collected. The larger receptive field size of the T4 cells is important; that should get mentioned in the text where this data is described.*

First of all, we enlarged the figures, added axis labels to the bar histograms and also visualized the entire receptive field as an additional 3D bar plot. This should help quite a bit. We also added the following explaining sentence to the description of the receptive field of T4:

"This indicates that T4 neurons pool excitatory synaptic input from more than one column, consistent with their morphology (Figure 1) and expected from a motion detector required to integrate information from spatially offset input. "

3) Figure 3 models. It was not clear to me what the modified models were. Specifically, do the models lacking null direction suppression or preferred direction enhancement still consist of three inputs (as in Figure 3) just with one arm disabled? Or are they models with two inputs only (in a more classic Reichardt configuration)? A schematic of these models added to Figure 3 could be helpful.

We now included schemata to illustrate the configuration of the 3 models as suggested.

*4) Figure 3. I am confused by the description of this figure. The legend does not have a panel d, but the legend for panel e suggests it is describing an array of model units, not a full T4 cell. That appears to make sense from comparing Figure 3 and d. But the main text describes a full model T4 cell. Because of these inconsistencies I could not evaluate that part of the paper.*

We use the term model units as a general term for any of the 3 models that we compare. Figure 3 (now 5d) compares the directional tuning of all 3 models, illustrating the sharpest tuning for the full model incorporating both PD enhancement and ND suppression. For clarity, we also included schemata of all 3 models. The typo in the legend was corrected from e to d.

*Reviewer #3:*

*In this tantalizing manuscript, Haag, Borst and colleagues explore the underlying basis of visual motion direction computations in the Drosophila optic lobe. In both the vertebrate retina and in the early visual system of flies, this computation has been studied in the framework of two classic models of visual motion direction: the Hassenstein-Reichardt (H-R) model and the Barlow-Levick (B-L) model. Mechanistically, both rely on combining input from spatially offset pathways, with input from an adjacent location temporally delayed relative to the central pathway. One (H-R) produces an excitatory "preferred-direction" (PD) response by multiplication with temporally delayed input from a preceding location, while the other (B-L) uses division by input from a temporally delayed location in the "null direction" (ND). Direct evidence for the latter has been obtained in direction-selective retinal ganglion cells (DSRGCs), where starburst amacrine cells provide the appropriate ND inhibition. In the Drosophila optic lobe, the H-R model has thus far been the mechanism most often proposed for direction selectivity for large field motion, with information from "ON" and "OFF" motion pathways represented by so-called T4 and T5 cells respectively being combined in vertical system (VS) and horizontal system (HS) lobula plate tangential cells (LPTCs). Recent anatomical and electron microscopy (EM) evidence has suggested a more complex picture at the level of the input to T4 and T5 neurons – their inputs comprise more cell types than previously believed. In this study, the authors use precise optical stimulation that takes into account the fly's neural superposition eye (Franceschini, 1975) along with two-photon calcium imaging to suggest that B-L-like inhibition may combine with H-R like excitation to shape the responses of T4 neurons. The results are interesting but the evidence provided is indirect and the picture still somewhat murky. Detailed comments follow.*

The major claim of the paper is that precise stimulation of single lamina columns reveals PD and ND responses that localize to different sub-regions of the T4 neuron dendrites. However, this claim, which is highlighted in the abstract, goes beyond the evidence presented, which is entirely based on recordings in T4 axons. Unless the authors can present direct anatomical and physiological evidence for such localized dendritic input, I would suggest that this claim be removed or significantly tempered and moved to Discussion.

We rephrased this statement in terms of offset receptive fields that are shown in Figure 3. Given the columnar organization of the input elements and the anatomy of the dendrite, we still find it highly likely that the inputs representing the enhancing and suppressing arm will be separated on the dendrite. This argument, however, is now moved to the Discussion.

Similarly, the B-L model proposes inhibition of one signal by another for ND motion, but the authors present no direct evidence for such inhibition. What is shown instead is supra- or sub-linear summation upon sequential (PD vs ND) excitation of successive lamina columns. Why does this have to necessarily result from inhibition in T4 dendrites rather than have presynaptic origins? While it is true that medulla neurons are not known to be sensitive to motion-direction, such neurons are known to have suppressed responses to bars wider than their receptive field (T5 input neurons for OFF pathway; Serbe et al., 2016 from the Borst lab), so might a skewed sampling of such inputs by T4 perhaps confer the observed ND suppression?

We precisely constructed this telescopic stimulation device to ensure accurate activation of single columns with minimal stimulation of the surround (see L2 imaging experiments in Figure 2) in perfect register with the columnar raster. This point is also illustrated in the recordings using the LED arena. Stimulation with larger bars or not in perfect register (see sharpness of the central peak for small stimuli) leads to a strongly reduced flicker response. However, surround inhibition of presynaptic elements would affect all stimuli regardless of their direction (single flicker, PD and ND) and thus cannot account for the ND suppression seen that is calculated with respect to the individual flicker responses.

Related to the above point, the model (Figure 3) predicts that we should be able to directly observe inhibition (not just suppression of a strong excitatory response), even if it is not large, when the 'wrong' lamina column is illuminated in a flicker experiment. This seems never to be observed in the experiments shown (Figure 1). What is seen instead is sub-linearity (or supra-linearity), which the authors call a non-linear response component. Is this a calcium indicator limitation even with the higher-baseline GCaMP6m? If there is inhibition, might it be easier to pick up directly in the dendrites of single T4 neuron or do the authors find this experimentally infeasible?

We did not observe an inhibition in terms of a decrease in the calcium signal reflecting a hyperpolarization. Instead, activation of the suppressive input strongly suppresses subsequent stimulation of the direct input, most likely via a shunting inhibition or a presynaptic inhibition of the suppressing input onto the direct input. We included an additional figure (Figure 3) illustrating that the suppression not just leads to smaller-than-the-sum-responses, but reduces the responses below the level of isolated center stimulation. The figure also shows that this suppression builds up along sequences of steps in the null direction. We do not think it likely that it would be possible to directly observe an inhibition/decrease in the calcium signal in specific subregions of the dendrite that is not visible in the axon terminal or in the neuron close to the lobula plate. Those of the small neurons tested so far in the optic lobe tend to be electrically compact (Yang et al., 2016).

In Figure 1, the authors state that their telescopic illumination allows them to illuminate only a single lamina column, but the lamina responses suggest that neighboring columns are also stimulated. In the manuscript (introduction section), these responses are described as 'negligible', but they seem to reach 50% of peak responses. This requires further discussion, because the study's results rely heavily on the high precision of optical illumination. If the responses reflect the limits of the illumination system, it would be useful to discuss how that might affect the interpretation of results. Also, on the same note, it would be great to see what a 'standard' localized stimulus presentation (i.e., targeting single ommatidia and sequences of ommatidia without using the telescopic refinement) would evoke in T4 neurons. This would highlight the importance of telescopic stimulus presentation for visual motion studies, which would be useful to the field.

To validate the alignment of the telescopic stimulation device we used a micrometer scale slide. When the first lens after the stimulation objective is taken out, the device changes from a telescope to a microscope. By putting a scale slide in the focal plane of the stimulation objective, we can image the micrometer scale with the CMOS-camera and position light at different positions on the scale. By using the objective used for 2-photon microscopy the positioning of the stimuli on the scale can be visualized with the CCD-camera included in the 2-photon setup. We then aligned the beam path of the stimulation device so that the position of the light stimuli on the CMOS-camera and the CCD-camera coincide.

We now also increased the number of recordings from L2 neurons and show that the stimulation is restricted to single columns, in the limits of the optics of the fly’s eye (below 30%). The responses to the outer ring of neuro-ommatidia around the central column are below 12% and indeed negligible, as stated in the manuscript. A comparison to classical stimulation is given by the LED arena experiments (former Figure 2, now Figure 4). Here we used large and small bars, at different positions limited by the resolution of the LED arena. Presumably inhibitory surrounds of the input elements lead to a strong reduction in the flicker responses of large bars as compared to small bars. With small bars, both a preferred direction enhancement and a null direction suppression could be revealed (Figure 4). However, in contrast to the telescopic stimulation, the register of the stimulus to the columnar raster has to be indirectly inferred from the flicker amplitude.

Although perhaps this is beyond the scope of this study, it would be very interesting to know what T5 neuron responses look like with telescopic stimulation. Do they also show B-L like ND suppression in the OFF pathway? Given that the authors have access to all the T5 inputs (Serbe et al., 2016), this could also provide an easier path to a more complete explanation in these neurons as opposed to T4, where the authors do not say much about what the potential inputs to these neurons might be and do not explore how they might act together to produce the observed response properties.

We are indeed planning to repeat this study on T5 cells and block their known input neurons in order to find out which neuron is doing what. This, however, will for sure take quite a long time to be completed.

*Lastly, it is still left unclear why T4 (and, perhaps T5) might need a potential B-L like mechanism in addition to H-R if there is no difference between the combination and pure H-R for LPTCs. Perhaps the authors could speculate on this in Discussion.*

This is an excellent point. We added the following text in the Discussion to address the potential functional consequences: "While the output of the full Hassenstein-Reichardt-correlator after the subtraction stage shows a high degree of direction-selectivity in the absence of null-direction suppression, the relatively small differences between poorly tuned signals would be very sensitive to noise. Improving the direction-tuning already at the level of the half-detectors by the additional null-direction suppression increases robustness to noise and might in addition be energetically less costly. “